# *Oudemansiella raphanipies* Polysaccharides Improve Lipid Metabolism Disorders in Murine High-Fat Diet-Induced Non-Alcoholic Fatty Liver Disease

**DOI:** 10.3390/nu14194092

**Published:** 2022-10-01

**Authors:** Haitao Jiang, Hua Zhu, Guangming Huo, Shengjie Li, Yulong Wu, Feng Zhou, Chun Hua, Qiuhui Hu

**Affiliations:** 1College of Food Science and Technology, Nanjing Agricultural University, No. 1 Weigang Road, Nanjing 210095, China; 2School of Food Science, Nanjing Xiaozhuang University, No. 3601 Hongjing Road, Nanjing 211171, China; 3School of Life Science and Chemical Engineering, Jiangsu Second Normal University, No. 77 Beijing West Road, Nanjing 210013, China

**Keywords:** *Oudemansiella raphanipies*, polysaccharides, non-alcoholic fatty liver disease, metabonomics

## Abstract

*Oudemansiella raphanipies*, also called “Edible Queen,” is a mushroom that possesses antioxidant, anti-inflammatory, anti-bacterial, anti-tumor and immunity-enhancing properties. The present study aimed to assess the effect of O. *raphanipies*-derived polysaccharide (ORPS) on the progression of nonalcoholic fatty liver disease (NAFLD) in mice. We studied the structure of ORPS-1 by high-performance gel permeation chromatography (HPGPC), ion chromatography-mass spectrometry (GC-MS), and Fourier transform-infrared spectroscopy (FT-IR). ORPS-1 mainly comprised galactose, fucose, glucose, mannose, and xylose, following an 18:6:6:4:1 molar ratio. In addition, the therapeutic effect as well as a potential mechanism of ORPS-1 in the treatment of high-fat diet (HFD)-induced NAFLD were investigated. The results showed that ORPS-1 improved liver function, ameliorated liver steatosis, and reduced lipid droplet accumulation in HFD mice. A metabolomics approach with GC-MS was utilized to evaluate liver improvement by ORPS-1 treatment. Principal component analysis showed that liver metabolic profiling was significantly altered by HFD feeding or treatment with an intermediate dose of ORPS-1 in mice compared with that of control mice. By investigating the metabolic pathways with identified biomarkers, various pathways such as steroid biosynthesis, valine, leucine, and isoleucine biosynthesis, glycerol phospholipid metabolism, glyceride metabolism, and arginine and proline metabolism in HFD mice were observed to be significantly influenced by ORPS-1 treatment. The results indicate ORPS-1 metabolic effects on liver tissues, provide methods for assessing the molecular impact of ORPS-1 on NAFLD, and suggest the potential mechanism underlying its health benefits.

## 1. Introduction

Nonalcoholic fatty liver disease (NAFLD), which is characterized by a significant increase in hepatic fat accumulation (steatosis), is the most frequent global liver disease, and its incidence continues to increase, especially in young adults and children [1]. Epidemiological studies have demonstrated that the global incidence of NAFLD is 25.24% [2]. There is growing evidence indicating that NAFLD is triggered by various factors such as age, sex, abnormal obesity, gut microbiota, and insulin resistance. NAFLD is associated with metabolic syndrome development as well as metabolic risk factors such as diabetes, obesity, and dyslipidemia [3]. Current treatment of NAFLD includes bariatric surgery and drug intervention, including thiazolidinediones. However, these methods can lead to various side effects, including increased weight and osteoporosis [4]. The etiology and pathogenesis of NAFLD are unclear. It has been shown that edible and medicinal fungi such as *Antrodia cinnamomea, Sarcodon aspratus, Amilariella mellea,* and *Lentinus edodes* impart certain therapeutic or inhibitory effects on the development of fatty liver to prevent or treat NAFLD, although it is necessary to identify their bioactive components and develop additional functional foods or dietary supplements [5,6].

Edible and medicinal fungi have been extensively cultivated and consumed globally, particularly in countries in Southeast Asia such as Japan, China, and South Korea. Edible mushrooms have been historically harvested and consumed or employed in traditional medicine as they are low in calories and taste delicious. Recent studies have suggested that mushrooms and their extracts may be potentially employed as therapeutics and prebiotics [7,8,9]. The major components of mushrooms eliciting prebiotic activity include indigestible polysaccharides such as lentinan and pleuran [10,11].

*Oudemansiella**raphanipies* is an edible mushroom that has high levels of amino acids, various biological enzymes vitamins, and alcohols, as well as a wide range of bioactive components, including polysaccharides, polyphenols, and flavonoids [12,13]. These active ingredients are often used in studies on antioxidation, anti-inflammation, lung protection, kidney protection, and carbon tetrachloride-induced hepatotoxicity in rats [14,15,16]. Among these, *O. raphanipies*-derived polysaccharide (ORPS) is the main active substance in *O raphanipies*, which has anti-inflammatory, antioxidant, immunity-enhancing, anti-tumor, and anti-bacterial properties [15,17]. However, to our knowledge, no systematic investigations on the inhibitory effect of ORPS on NAFLD development have been conducted and the underlying molecular mechanisms remain unclear.

Here, we purified the ORPS and designated this as ORPS-1. A high-fat diet (HFD)-induced NAFLD mouse model was utilized to assess the benefits of applying ORPS-1 to NAFLD. A metabolomics methodology using gas chromatography-mass spectrometry (GC-MS) was employed to elucidate the potential molecular mechanism underlying ORPS-1 in relation to liver improvement. This study provides new insights into evaluating the functional components of natural macromolecules and corresponding methods.

## 2. Materials and Methods

### 2.1. Isolation and Purification of Polysaccharides

*O. raphanipies* was obtained from the Institute of Mycology, Xiaozhuang University, Nanjing, China. We prepared *O. raphanipies* powder using water and centrifugation at 5000 rpm for 10 min, deproteinization by the Sevage method [18], and then freeze-drying to yield crude polysaccharides. Then, the crude polysaccharides were processed with Diethylaminoethyl-52 cellulose (DEAE-52) and using Sephadex G-200 (2.6 cm × 60 cm) columns (Shanghai Kaiyang Biotechnology Co., Ltd., Shanghai, China) to generate purified ORPS.

### 2.2. Characterization of Monosaccharides

Due to the complex composition of polysaccharides, the monosaccharide composition was determined via ion chromatography. The purified polysaccharides (5 mg) were accurately weighed, placed into an ampoule for hydrolysis, and then blow-dried using nitrogen gas flow. Water was later added, followed by vortex mixing, and then the mixture was centrifuged at 12,000 rpm for 5 min. The hydrolysate was then passed through an ICS5000 column (Thermo Fisher Scientific, Waltham, MA, USA) that was equipped with an electrochemical detector.

### 2.3. Fourier Transform-Infrared (FT-IR) Spectroscopy

After drying, approximately 2 mg of samples were ground in 200 mg of spectroscopic-grade potassium bromide powder and then pressed into a 1-mm pellet. We recorded the FT-IR spectra with an FT-IR spectrophotometer (Nicolet IR200, Nicolet Co., Madison, WI, USA) at a frequency range of 4000–400 cm^−1^.

### 2.4. Determination of Molecular Weight (Mw)

We determined the Mw and purity of the polysaccharides by high-performance gel permeation chromatography (HPGPC). The sample was accurately weighed, the solution was prepared and centrifuged, the supernatant was filtered, and finally placed in the injection vial until analysis. The following conditions were used: chromatographic column: BRT105-104-102 series gel column (8 × 300 mm, BoRui Saccharide, Yangzhou, China); flow rate: 0.6 mL/min; mobile phase: 0.05 mol/mL NaCl solution; injection volume: 20 mL; column temperature: 40 °C; and detector: differential detector RI-10A (Shimadzu, Japan).

### 2.5. Development of NAFLD Mouse Model

All animal experiments were approved for use by the Animal Care and Use Committee of Nanjing Agricultural University (SYSK2021-0086). Male mice (C57BL/6J; 4-week-old; weight: 20 ± 2 g) were obtained from the Nanjing Qinglongshan Animal Farm (License number: SCXK2017-0001). After acclimatization in the laboratory for three days, the C57BL/6J mice (*n* = 80) were randomly divided into two groups, namely, a control group (CON group, *n* = 15) and an HFD group (HFD, *n* = 65). During the experiment, mice in the CON group were fed a normal pellet feed(corn meal(25%) + soybean meal(20%) + bran(18%) + flour(20%)+ yeast(1%)+ milk powder(1%) + bone meal(1%) + rice(4%)+ fish meal(1%) + salt(5%) + composite additives (FeSO_4_ (7H_2_O), MnSO_4_ (H_2_O), CuSO_4_ (5H_2_O), AlCl_3_ (5H_2_O) and Zn SO_4_ (7H_2_O))(1%)), while those in the HFD group were fed high-fat feed(lard oil(10%)+ egg yolk powder(10%)+ cholesterol(1%)+ bile salt(0.2%) and basic feed(78.8%)). After 12 weeks, three mice in each group were randomly selected for analyzing serum biochemical indicators and liver pathology tests. Total cholesterol (TC), triglycerides (TG), and low-density lipoprotein cholesterol (LDL-C) indicators of blood lipids were significantly increased, and more than 1/3 of the cells in the liver showed steatosis; therefore, the model of NAFLD was considered successful [19].

The mice model group was randomly assigned to one of four groups (10 mice/group). One group was administered intragastric normal saline (HFD + SP, model group), and three groups were administered ORPS-1 at different doses (low dose-receiving group (LORPS-1), HFD + 50 mg/kg ORPS-1; medium dose-receiving group (MORPS-1), 100 mg/kg ORPS-1; and high dose-receiving group (HORPS-1), 200 mg/kg ORPS-1), and gavage was performed every day for four weeks.

The entire experiment was conducted for a total of 16 weeks, with body weights measured every week. Upon completion of the experiment, the mice fasted overnight. Afterward, the mice were anesthetized (intraperitoneal injection of 4% pentobarbital sodium (45 mg/kg body weight)) and blood samples were collected from the orbital plexus from which serum was collected by rapid centrifugation (3500 rpm for 10 min at 4 °C) and subsequently stored at −80 °C until analysis. After the exsanguination, the mice were sacrificed by cervical dislocation. If necessary, the animals were re-anesthetized by pentobarbital sodium. The mice remained anesthetized throughout the operation, and then the breathing and heartbeat of the mice were monitored in order to ensure their actual death. The liver tissues were immediately separated, weighed, and then stored at −80 °C for further analysis. All the procedures were performed in accordance with the published guidelines of the China Council on Animal Care.

### 2.6. Biochemical Assays

The levels of serum lipids such as TG and TC, as well as hepatotoxicity biochemical indicators such as LDL-C, alanine transaminase (ALT), aspartate aminotransferase (AST), and high-density lipoprotein cholesterol (HDL-C), were measured with an AU5800 automatic analyzer (Beckman Coulter, Inc., Brea, CA, USA).

The liver tissues were weighed according to the weight (g): volume (mL) = 1:9 and added to 9 times the volume of homogenate medium accurately. Then, the mixture was homogenized mechanically under ice water. The samples were centrifuged at 3000 rpm for 10 min and the supernatant was collected. The activities of hepatic TC and TG, as well as superoxide dismutase (SOD), glutathione peroxidase (GSH-Px), malondialdehyde (MDA), total antioxidative capacity (TAOC), are considered indicators of the antioxidant status of liver tissues, were evaluated using commercial kits (Nanjing Jiancheng Biology Engineering Institute, Nanjing, China), and normalized to albumin (BSA) reference levels. Protein concentration was measured by Bradford’s method [20].

### 2.7. Liver Histopathological Examination

The mice from each group were euthanized, then their livers were isolated and weighed, followed by a calculation of liver indices (liver index (%)  =  liver wet weight/mouse body weight × 100%). Tissue samples from the largest liver lobe (i.e., left side of the liver) were fixed with 10% formalin solution, embedded in paraffin, stained using hematoxylin and eosin, and then examined under a light microscope. Another liver tissue slice was fixed with 2.5% glutaraldehyde and observed under an electron microscope (JEM-2000EX, JEOL, Ltd., Duzhaodao City, Japan).

### 2.8. Metabolite Extraction

Liver tissues were placed in a test tube, to which a pre-cooled extraction mixture (3:1 methanol/chloroform) and an internal standard solution (adonitol, 0.5 mg/mL raw solution, Sigma-Aldrich, Saint Louis, MO, USA) were added and vortexed, subjected to ultrasonication in ice water, and centrifuged, with the supernatant was transferred into a fresh tube to prepare quality control (QC) samples. Upon evaporation in a vacuum concentrator, 50 μL of methoxyamination hydrochloride were added and derivatized by N, O-*Bis*(trimethylsilyl) trifluoroacetamide reagent (1% trimethylsilyl chloride, *v*/*v*). Approximately 5 μL of fatty acid methyl esters were added to the QC sample. All samples were subsequently analyzed on an Agilent 7890 gas chromatograph (Agilent Technologies, Santa Clara, CA, USA) that was coupled with a time-of-flight mass spectrometer.

### 2.9. Statistical Analysis

The data were expressed as the mean ± standard deviation. Statistical significance among various experimental groups was assessed via one-way ANOVA by the Bonferroni posthoc test (GraphPad Prism version 7.0; GraphPad Software, San Diego, CA, USA). Differences with *p* < 0.05 were deemed significant. Raw data analysis such as peak extraction, baseline adjustment, alignment, deconvolution, and integration was performed using Chroma TOF software (Ver 4.3x, Laboratory Equipment Corp., St. Joseph, MI, USA). In addition, the LECO-Fiehn Rtx5 database was employed for metabolite identification via matching the mass spectrum and retention index. We removed peaks that were detected in less than half of the QC samples or with a relative standard deviation > 30% [21].

## 3. Results and Discussion

### 3.1. Primary Structure of ORPS-1

The biological activity of polysaccharides mainly depends on their structure, such as chemical composition, molecular weight (Mw), sugar chain position, and linkage [22]. The Mw of ORPS was determined using HPGPC (Figure 1A). The symmetrical elution peaks of ORPS were observed, indicating that the Mw distribution of the polysaccharide was relatively concentrated. In this study, we purified the main polysaccharide, ORPS-1, for further analysis. Based on the standard curve, the average Mw of ORPS-1 was 24.9 kDa. Figure 1B illustrates the FT-IR spectrum of ORPS-1 within the absorption range of 4000–400 cm^−^^1^. We observed a stretching vibration absorption peak at 3388 cm^−1^, which we ascribed to the O-H bands. An absorption peak was detected at 2925 cm^−1^ and may be attributable to C-H stretching vibration. The absorption peaks identified at 1716 cm^−1^ and 1538 cm^−1^ could be ascribed to C=O stretching vibration. An absorption peak detected at 1643 cm^−1^ could be attributable to crystalline water. The absorption peaks at 1417 cm^−1^, 1149 cm^−1^, and 1072 cm^−1^ may be attributed to C-O stretching vibration. The absorption peaks at 1247 cm^−1^ and 1027 cm^−1^ may be ascribed to O-H variable angle vibration. An absorption peak detected at 971 cm^−1^ may be attributable to the rolling vibration of methylene at the end of the pyran ring. An absorption peak observed at 919 cm^−1^ was ascribed to the asymmetric ring stretching vibration of the pyran ring. An absorption peak detected at 765 cm^−1^ was attributable to the telescopic vibration of the symmetrical ring of the pyran ring [23,24]. Overall, these results demonstrated that ORPS may be pyran-type polysaccharides.

The monosaccharide standard and the results of ion spectrometric analysis are shown in Figure 1C,D, respectively. According to the retention time of the monosaccharide standard (Figure 1C), we report that ORPS-1 comprised galactose, fucose, glucose, mannose, and xylose at a molar ratio of 18:6:6:4:1 (Figure 1D). Among these components, galactose accounted for the largest proportion. Several studies have shown that polysaccharides containing fucose show a relatively higher liver protection activity when studying the liver protection activity of fungal polysaccharides [25,26]. Therefore, fucose in ORPS-1 may play an important role in liver protection. Overall, these findings lay the foundation for the structural determination of ORPS-1, assessment of its structure-activity relationship, and the development of pharmacological agents against NAFLD and other diseases.

### 3.2. ORPS-1 Improved Liver Function in HFD Mice

In our animal experiment, the effects of ORPS-1 on liver indices and body weight of NAFLD mice were investigated. Body weight and liver index are generally used to assess the pathological conditions of animals. Liver index is an important indicator in the evaluation of liver poisoning. At the start of the ORPS-1 treatment, the HFD mice were divided into four groups, with no apparent differences in initial average body weight (Figure 2). Compared with the HFD group, the weight of mice in the MORPS-1 and HORPS-1 groups slowly increased and later significantly decreased after four weeks of treatment. Furthermore, the liver indices were also reduced in the HFD group treated with MORPS-1 relative to that the HFD group that did not receive treatment. The changes induced by HFD in the body weight and liver index of the mice were reduced by ORPS-1 treatment, indicating that ORPS-1 can alleviate liver damage in NAFLD mice.

In addition, we further explored the effects of ORPS-1 treatment on levels of serum and liver biochemical indicators and liver morphology in NAFLD mice. AST and ALT are generally expressed in liver cells and serve as indicators of liver damage [27]. Under normal metabolic conditions, serum AST and ALT levels are low; when liver tissues are damaged, their cell membrane permeability changes, and a large number of serum enzymes enter the blood circulation and the activity of these enzymes increases. In the present study, serum AST and ALT activity in NAFLD mice significantly increased, whereas these decreased in the ORPS-1 treatment group (Figure 3A,B), which is consistent with previous reports [20].

At the same time, a direct relationship between liver function and blood lipid level exists. Blood lipids are mainly evaluated using serum TC, TG, LDL-C, and HDL-C levels [7]. Cholesterol is a fat-soluble steroid and is essential for cell membrane formation in the human body. It can be transported to the liver through blood and is metabolized by carriers such as LDL-C and HDL-C. Impairment of liver function results in abnormal liver lipid metabolism, cholesterol degradation is blocked, and blood lipids levels increase. Studies have shown that chronic liver injury is associated with abnormal blood lipid levels in the body [28]. In the present study, HFD treatment induced an increase in serum TC, TG, and LDL-C levels, a reduction in HDL-C levels in NAFLD mice (Figure 3C–F), and disruption of blood lipid metabolism, which are concordant to previous findings. This abnormal change was reversed by ORPS-1 treatment, indicating that ORPS-1 can repair liver damage to regulate blood lipid metabolism [20]. In normal physiological settings, the antioxidant system maintains the production and removal of free radicals at a steady state. Upon disruption of this balance, various free radicals are not processed, resulting in oxidative stress. The level of free radicals and lipid peroxide status in liver tissues can be assessed based on antioxidant enzyme (SOD, GSH-Px, and TAOC) activity and lipid peroxide (MDA) content. Oxidative damage in the gallbladder is reported to be directly related to the free radicals produced in the process of lipid peroxidation [20]. Therefore, inhibition of oxidative stress generation can protect the liver. In this study, compared with the CON group, hepatic TC, TG, and MDA levels had significantly increased, whereas SOD, GSH-Px, and TAOC levels were reduced in the HFD group (Figure 4). In particular, MORPS-1 and HORPS-1 treatments enhanced hepatic SOD, GSH-PX, and TAOC levels and reduced liver MDA levels (Figure 4). These results indicate that ORPS-1 has high antioxidant activity and radical-scavenging activity that in turn alleviates liver oxidative damage.

### 3.3. Histological Analysis Shows That ORPS-1 Treatment Ameliorates Liver Injury

Histological analysis revealed normal liver sections in the CON group as depicted by prominent nuclei and nucleoli, well-preserved cytoplasm, and distinct central veins (Figure 5A). The HFD induced extensive liver damage, which is characterized by hepatocyte necrosis, moderate to severe cell degeneration, and lipid droplet accumulation (Figure 5B). LORPS-1, MORPS-1, and HORPS-1 treatments ameliorated liver damage of HFD mice to varying degrees by turning bullous steatosis into microvesicular steatosis and reducing steatosis (Figure 5C–E), especially MORPS-1 treatment. Electron microscopy analysis showed marked steatosis and vacuolization in hepatocytes of HFD mice, but not from CON mice (Figure 6A). MORPS-1 and HORPS-1 treatment ameliorated vesicular steatosis and reduced lipid accumulation in hepatocytes of HFD mice (Figure 6C–E). In general, the biochemical indices of serum and liver and the histological evidence show that MORPS-1 treatment imparts preventive and therapeutic effects on liver injury in NAFLD mice. Furthermore, we assessed mouse sera from different treatment groups using metabolomics techniques trying to elucidate underlying mechanisms at the metabolic level.

### 3.4. Metabolic Profiling of Mice Liver Tissue

To investigate the influence of ORPS-1 treatment on liver metabolites, the liver samples from the mice of CON, HFD, and MORPS-1 groups were analyzed using GC-MS. The QC samples were concentrated in the principal component analysis (PCA) diagram (Figure 7), indicating that the established method may be employed in identifying various metabolites in different individual samples [29]. The score chart in Figure 7A shows the trend of separating the among classes. These results imply that the CON, HFD, and MORPS-1 samples are distributed in different groups based on the first two principal components, whereas there were no outlier samples. This suggests that there are metabolic differences among the CON and HFD and MORPS-1 groups (Figure 7A).

To clearly distinguish the experimental groups and increase the validity and reliability of the results, we employed an OPLS-DA to accurately assess the metabolic patterns of our samples. The quality of the model was investigated by R^2^ and Q^2^ values. R^2^, which represents the proportion of variance, separately explained and predicted the PCA model. In Figure 7B,C, the OPLS-DA parameters of the model were expressed by an R^2^ of 0.81 and Q^2^ of −0.88 while comparing CON and HFD groups, and the R^2^ of 0.87 and Q^2^ of −0.68 while comparing HFD and MORPS-1 groups. These parameters depicted preferable model stability and predictability as well as effectively revealed the metabolic differences between the CON and HFD groups, as well as the HFD and MORPS-1 groups. The permutation test involving orthogonal projections to latent structures discriminant analysis (OPLS-DA) showed greater accuracy than PCA analysis in classifying normal and HFD mice (Figure 7B,C).

### 3.5. Analysis of Total Metabolites

Figure 8A shows that relative to the CON group, 227 metabolites with different levels such as 132 upregulated metabolites and 95 downregulated metabolites could be identified in hepatic tissues from the HFD group. Figure 8B shows that relative to the HFD group, 102 metabolites with different levels such as 50 upregulated metabolites and 52 downregulated metabolites were observed in liver tissues from the MORPS-1 group. Further analysis showed that after the action of MORPS-1, a considerable part of the metabolites returned to their initial expression levels before the induction of a high-fat diet, indicating that the MORPS-1 group alleviated liver injury in NAFLD mice to a certain extent.

By clustering all detected metabolites, the metabolites of 10 samples from each group were clumped into one group (Figure 8C–E), implying that the metabolic patterns of these samples were generally similar after salt treatment. In addition, the CON, HFD, and MORPS-1 groups exhibited regions with relatively clear high and low expression, whereas the high and low expression regions in the HFD group were discordant to those of the CON and MORPS-1 groups, indicating that MORPS-1 treatment imparted a significant effect on liver metabolism in HFD mice.

Figure 8C shows that relative to the control group, 81 metabolites exhibited different levels in the model group, with 43 that were upregulated and 38 downregulated. Compared with the model group, 41 metabolites with different levels were identified, 21 of the differential metabolites in the MORPS-1 group were upregulated, and the remaining 20 were downregulated (Figure 8D). The results showed that HFD led to significant changes in many metabolites in mouse livers. Overall, 24 biomarkers were observed in the three groups, of which six were deemed upregulated and 18 downregulated in the model group. The expression of these metabolites in the MOPRS-1 and control groups were similar, and the trend was opposite of that observed in the model group. These metabolites are strongly associated with biological processes such as energy metabolism, lipid metabolism, amino acid metabolism, and primary bile acid biosynthesis (Figure 8E).

In this study, enrichment analysis and topological analysis were conducted on the pathways of different metabolites, and the pathways were screened to identify key pathways with the highest correlation. Pathway analysis of liver samples is represented in the form of a bubble diagram. Figure 8F shows that relative to the control group, the metabolites observed in the model group are mainly associated with 23 metabolic pathways. Among these, the main metabolic pathways were related to linoleic acid metabolism, arginine and proline metabolism, pantothenic acid and coenzyme A biosynthesis, taurine and sub taurine metabolism, niacin and nicotinamide metabolism, and valine, leucine, and isoleucine biosynthesis. Figure 8G shows that relative to the model group, the differential metabolites in the MOPRS-1 group were mainly associated with 15 metabolic pathways. Of these, the main metabolic pathways were related to steroid biosynthesis, valine, leucine and isoleucine biosynthesis, glycerol phospholipid metabolism, arginine and proline biosynthesis, and glyceride metabolism. Steroid biosynthesis is associated with NAFLD development. Studies had shown that natural compounds could play a protective role against NAFLD development by regulating lipid metabolism and steroid biosynthesis. MORPS-1 imparted hepatoprotective effects in HFD-induced NAFLD mice [20].

In addition, amino acid metabolism is particularly sensitive to environmental changes [30,31]. Among the various metabolites, the biosynthesis of valine, leucine, and isoleucine is important in the development of metabolic diseases [32,33]. Changes in their contents might lead to oxidative stress generation in NAFLD [34]. Additionally, phosphate is an important component of the phospholipid bilayer, and an activator of many protein kinases that are regulated by various signal transduction pathways in mice and induce NAFLD development [35,36]. Hence, alterations in amino acid and free phosphate contents may have resulted from an inherent stress mechanism in mice. Notably, arginine, as a precursor of the bioactive factor nitric oxide, has a functional role and is associated with the mechanism of obesity control [35,36]. For example, L-arginine supplementation could reduce obesity, increase muscle mass, promote fat decomposition in adipose tissues, and effectively ameliorate the obesity crisis [13]. As mentioned above, these results confirmed that medium dose of ORPS-1 significantly affected amino acid-and steroid-related metabolic pathways and the lipid metabolism in NAFLD mice. Considering the stress response mechanism detected in the majority of mice, an intermediate dose of ORPS-1 could regulate amino acid or protein-related metabolic pathways to impart therapeutic effects on NAFLD.

## 4. Conclusions

In this study, we assessed the body weight, histopathology, and serum and liver biochemical indexes such as TC and TG in mice to determine whether the NAFLD mouse model was successful and preliminarily evaluated the therapeutic effect of ORPS-1 on NAFLD progression. We then analyzed the effects of different metabolites in NAFLD mice via liver GC/MS metabonomic technology. The levels of different hepatic metabolites in the HFD group were altered compared to those in the CON group, and the majority of these metabolic changes improved after treatment with an intermediate dose of ORPS-1. NAFLD lead to the development of lipid, glucose, and amino acid metabolism disorders due to changes in various metabolic pathways and body dysfunction. Metabolic pathway analysis showed that medium dose of ORPS-1 regulated the biological pathways and processes in NAFLD mice. These improvements disrupt the development of metabolic disorders to some extent and ultimately improves various physiological indexes in NAFLD mice. In conclusion, this study investigated the protective effect of ORPS-1 on mice with HFD-induced NAFLD. There is a need to elucidate the protective mechanisms of ORPS-1 in lipid metabolism disorders using hepatic proteomics and metagenomics of intestinal microflora.

## Figures and Tables

**Figure 1 nutrients-14-04092-f001:**
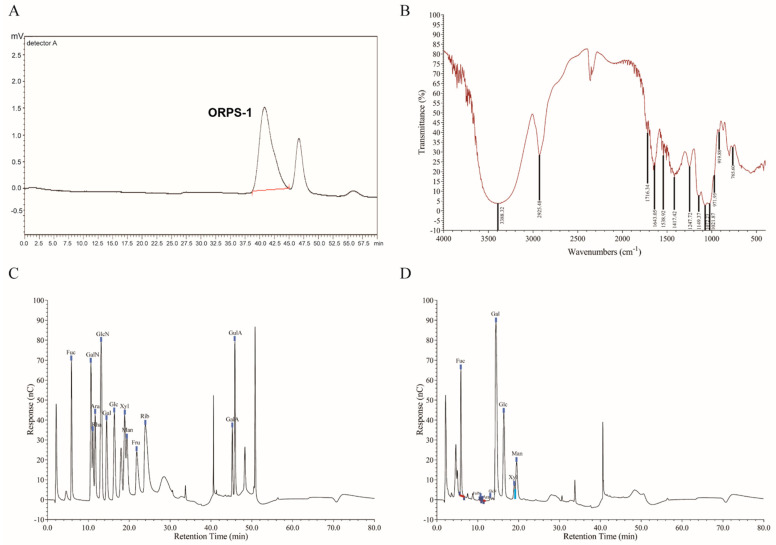
Primary structure analysis of ORPS−1(ORPS, *Oudemansiella raphanipies* polysaccharide). (**A**) HPGPC chromatogram of relative molecular weight of ORPS−1; (**B**) infrared spectroscopy analysis of ORPS−1; (**C**) Ion chromatography of standard monosaccharides and (**D**) Ion chromatography of ORPS−1. HPGPC, high-performance gel permeation chromatography; Ara, arabinose; GalA, galacturonic acid; GalN, galactosamine hydrochloride; GlcA, glucuronic acid; GulA, guluronic acid; Fuc, fucose; Fru, fructose; Rha, rhamnose; Gal, galactose; GlcN, glucosamine hydrochloride; Glc, glucose; GlcNAc, N−acetyl−D−glucosamine; Man, mannose; ManA, mannuronic acid; Rib, ribose; Xyl, xylose.

**Figure 2 nutrients-14-04092-f002:**
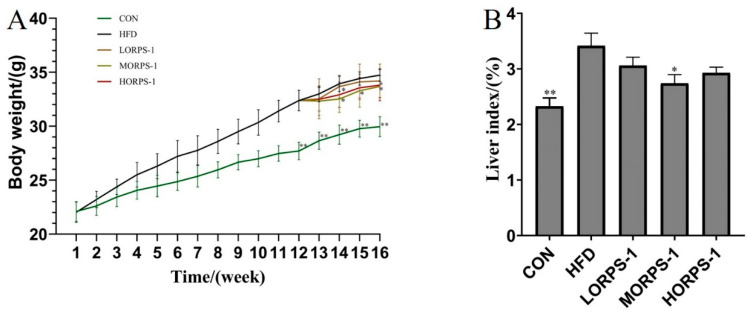
Effect of ORPS-1 (ORPS, Oudemansiella raphanipies polysaccharide) on the body weight (**A**) and various liver indices (**B**) of mice with NAFLD mice with nonalcoholic fatty liver disease. ORPS, *Oudemansiella raphanipies* polysaccharide; HFD, high-fat diet; CON, control group. * *p* < 0.05, vs. HFD group; ** *p* < 0.01, vs. HFD group. (LORPS-1, low dose-receiving group); (MORPS-1, medium dose-receiving group); (HORPS-1, high dose-receiving group).

**Figure 3 nutrients-14-04092-f003:**
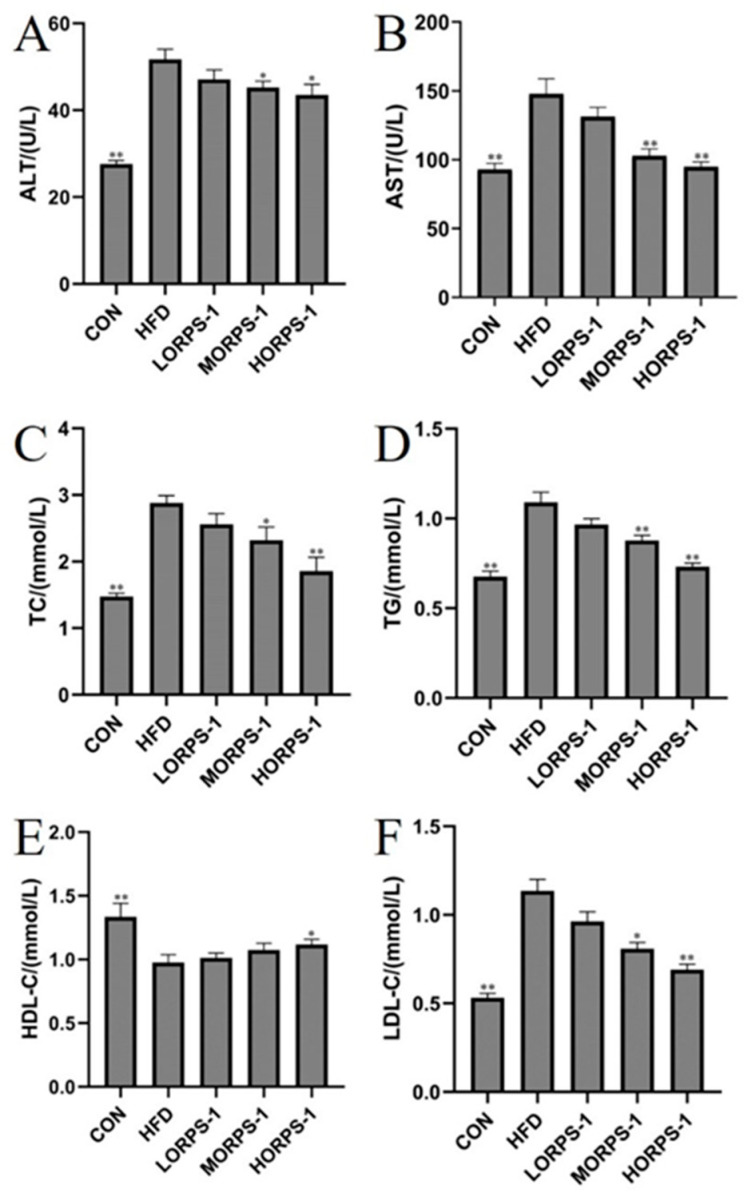
Effects of 16 weeks of ORPS-1 treatment on major serum indices in NAFLD mice. (**A**) ALT, (**B**) AST, (**C**) TC, (**D**) TG, (**E**) HDL-C, (**F**) LDL-C. Values are expressed as the mean ± SD. * *p* < 0.05, vs. HFD group; ** *p* < 0.01, vs. HFD group (*N* = 10 mice/group). ALT, alanine transaminase; AST, aspartate aminotransferase; HDL-C, high density lipoprotein cholesterol; LDL-C, low density lipoprotein cholesterol; TC, total cholesterol; TG, triglyceride.

**Figure 4 nutrients-14-04092-f004:**
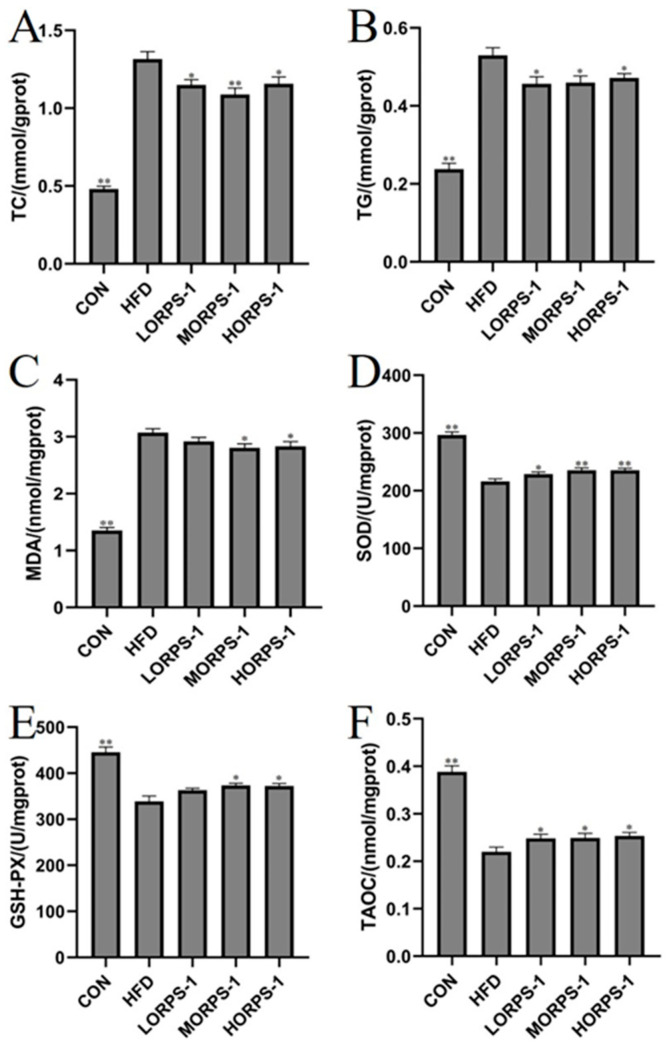
Effects of ORPS-1 (ORPS, Oudemansiella raphanipies polysaccharide) treatment on major hepatic indices in NAFLD mice. Effects of ORPS on liver metabolites. (**A**) TC, (**B**) TG, (**C**) MDA, (**D**) SOD, (**E**) GSH-PX, (**F**) TAOC. GSH-PX, glutathione peroxidase; MDA, malondialdehyde; TC, total cholesterol; TG, triglyceride; SOD, superoxide dismutase; TAOC, total antioxidative capacity. Values are expressed as the mean ± SD. * *p* < 0.05, vs. HFD group; ** *p* < 0.01, vs. HFD group. (*n* = 10 mice/group).

**Figure 5 nutrients-14-04092-f005:**
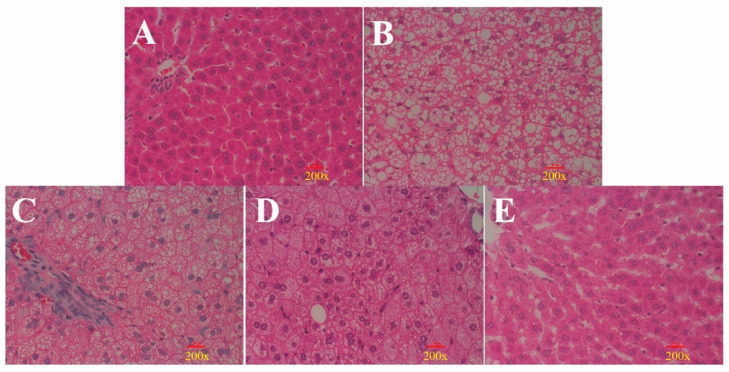
Effects of ORPS-1 (ORPS, Oudemansiella raphanipies polysaccharide) treatment on liver tissues observed via morphological analysis (hematoxylin and eosin staining, magnification 200×, microstructure). (**A**) normal control group; (**B**) nonalcoholic fatty liver disease model group; (**C**) LOPRS-1, 50 mg/kg ORPS-1 + HFD; (**D**) MOPRS-1, 100 mg/kg ORPS-1 + HFD; (**E**) HOPRS-1, 200 mg/kg ORPS-1 + HFD. ORPS, *Oudemansiella raphanipies* polysaccharide; HFD, high-fat diet.

**Figure 6 nutrients-14-04092-f006:**
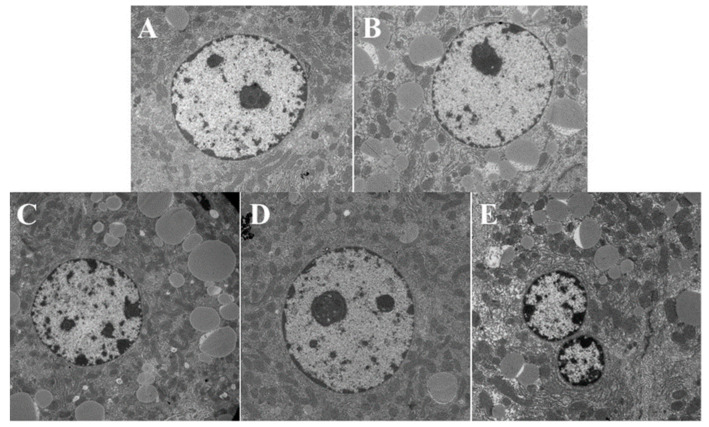
Effect of ORPS-1 (ORPS, Oudemansiella raphanipies polysaccharide) treatment on liver tissues in NAFLD mice (10,000× ultrastructure). (**A**) normal control group; (**B**) nonalcoholic fatty liver disease model group; (**C**) LOPRS-1, 50 mg/kg ORPS-1 + HFD; (**D**) MOPRS-1, 100 mg/kg ORPS-1 + HFD; (**E**) HOPRS-1, 200 mg/kg ORPS-1 + HFD. ORPS, *Oudemansiella raphanipies* polysaccharide; HFD, high-fat diet.

**Figure 7 nutrients-14-04092-f007:**
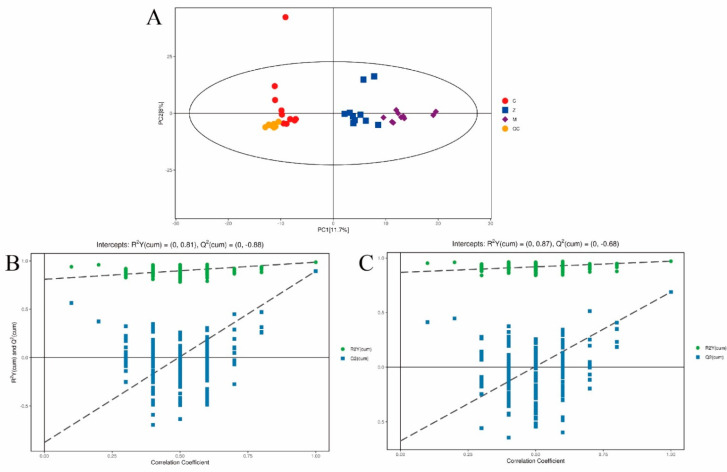
Score scatter plot for PCA model total with QC (Quality control). Global map of liver metabolites performed on a gas chromatography/mass spectrometry metabonomic platform. (**A**) Score scatter plot of the PCA model total with QC. Permutation test of orthogonal projections of latent structures discriminant analysis model for (**B**) group M vs. C and (**C**) group Z vs. M. C: Control group, M: high−fat diet group, and Z: MORPS−1 group (*n* = 10).

**Figure 8 nutrients-14-04092-f008:**
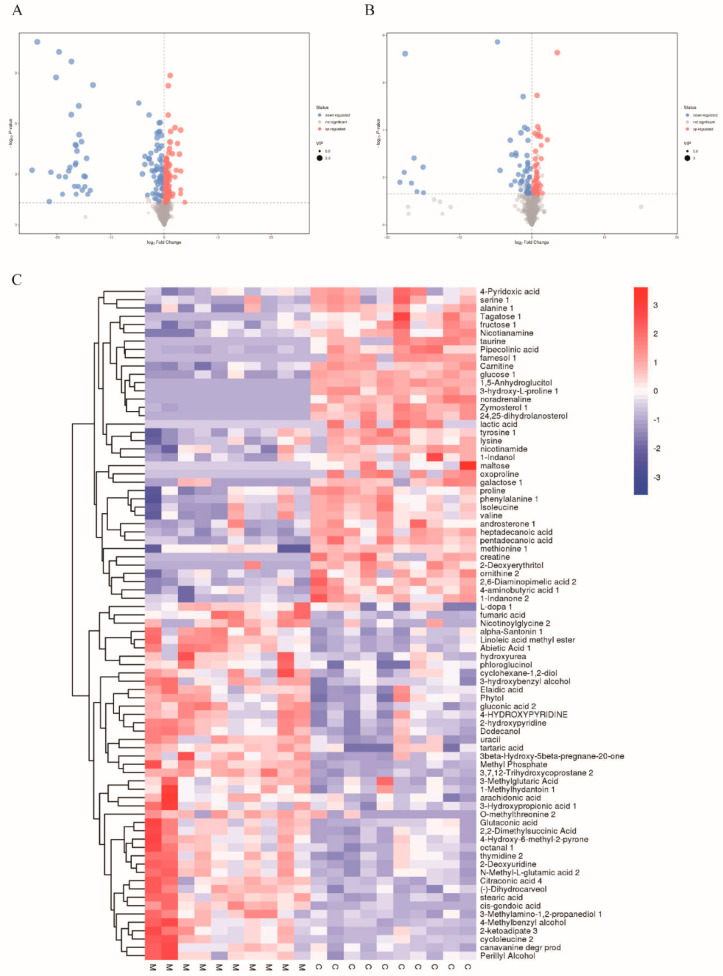
Volcano plot analysis for (**A**) group M vs. C and (**B**) group Z vs. M. Heatmap of hierarchical clustering analysis for (**C**) group M vs. C and (**D**) group Z vs. M. Heatmap of hierarchical clustering analysis among (**E**) group M vs. C and Z. Pathway analysis for (**F**) group M vs. C and (**G**) group Z vs. M.

## Data Availability

The data supporting the research for this study are available within the manuscript.

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
