# Peer review of "Oudemansiella raphanipies Polysaccharides Improve Lipid Metabolism Disorders in Murine High-Fat Diet-Induced Non-Alcoholic Fatty Liver Disease"

_nutrients, 2022, doi:10.3390/nu14194092_

Round 1

Reviewer 1 Report

From a scientific side, there is not much I have to add to the manuscript. The experiments and conclusions are sound and the work done by the authors fits into the chosen special issue of Nutrients.

However, the manuscript lacks a coherent formatting, with text sizes constantly changing. Furthermore, figures 1, 7 and 8 are of low quality and too small to read the given information. The authors need to adress these problems to achieve a good quality of presentation, otherwise the publication should not accepted. I suggest the authors to split up some of the figures, as often several smaller figures are put-together to a bigger one. This should likely lead to better visibility of the important information.

Reviewer 2 Report

Introduction

Line 63: The statement is not correctly cited. For example, carbon tetrachloride induced hepatotoxicity statement does not match with cited reference.

Method:

Line 113: Please mention the age of mice used.

Please mention the dietary contents of control diet and HFD.

Line 121: Please define TC, TG and LDL when first mentioned. This issue should be resolved throughout the manuscript.

Line 145. Please provide the method of lipid measurement in the liver tissue. Mention how the lipid was extracted from the hepatic tissue.

Result

The authors have provided the liver index data, however, there is no information about how this has been calculated. Please provide this information in method section in detail.

Figure 5. The scale bar is not clear in the histology figure. Please make sure the scale bar is clear in the figure. Also, was the histological slides evaluated by blinded investigators? 

Figure 8. The figure is hard to observe. Please provide high resolution figure with clear labels.

There are typographical error and different fonts size throughout the manuscript which should be resolved.
